# GLP-1 activates $K_{ATP}$ channels in coronary pericytes as the effector of brain-gut-heart signalling mediating cardioprotection

Svetlana Mastitskaya [1,3] ✉, Felipe Santos Simões de Freitas[2], Lowri E. Evans[1] & David Attwell [2,3] ✉

Failure to reperfuse the coronary microvasculature ("no-reflow") affects up to 50% of patients after unblocking a coronary artery that was causing ischaemia and acute myocardial infarction. This "no-reflow" is associated with reduced left ventricular ejection fraction, increased infarct size and death. We show that the incretin hormone GLP-1 (glucagon-like peptide 1) can be used to protect the heart after ischaemia by activating ATP-sensitive $K^+$ channels on pericytes that constrict coronary capillaries. Coronary capillary dilation can be activated pharmacologically or by vagally-mediated GLP-1 release from the gut evoked by skeletal muscle ischaemia, and is abolished by block or genetic deletion of pericyte $K_{ATP}$ channels. These results define a brain-gut-heart pathway mediating cardioprotection and suggest pharmacological therapies to reduce ischaemia-induced coronary no-reflow and improve post-infarct recovery.

Partial or complete occlusion of a coronary artery, often following atherosclerotic plaque rupture, leads to cardiac ischaemia with ST-segment elevation on the ECG. Primary percutaneous coronary intervention (PPCI) is the preferred treatment and aims to restore blood flow in the affected artery and minimise damage to the heart muscle. However, this does not guarantee reperfusion of the downstream capillaries supplying the myocardium[1,2] and a lack of capillary reperfusion—'no-reflow'—affects up to 50% of patients[3,4].

Infarct size (the amount of damaged heart tissue) is a predictor of adverse events and left ventricular remodelling after a heart attack. Cardioprotection studies frequently focus on reducing infarct size, but recent research suggests that the presence of no-reflow is an independent predictor of adverse outcomes and may be more significant than infarct size itself[4]. However, while many studies have examined infarct size, only a few have specifically addressed the microvascular occlusion causing no-reflow as a therapeutic target[5,6]. This is despite a meta-analysis[7] demonstrating a robust relationship between microvessel occlusion and mortality or hospitalisation for subsequent heart failure within 1 year, with a 1% increase in vessel occlusion predicting 14

and 11% increases in death and hospitalisation, respectively. Understanding and targeting therapeutically the mechanism(s) leading to vessel occlusion may therefore significantly improve outcome after cardiac ischaemia.

We have previously shown that pericyte-mediated capillary constriction contributes significantly to no-reflow after cardiac ischaemia[8]. For the brain, where the same mechanism operates, this results from a rise of intracellular calcium concentration, $[Ca^{2+}]$, triggering pericyte contraction during ischaemia, with subsequent pericyte dysfunction or death maintaining the constriction when the arterial blood supply is restored[9]. Similar events occur following renal ischaemia[10]. Despite the critical role of pericyte-mediated capillary constriction not yet being broadly known in the cardiac community[11], this finding has been independently reproduced for the heart[12]. We have shown that pericyte-mediated no-reflow can be significantly reduced by infusing adenosine at the time of reperfusion[8], and it has also been suggested that remote ischaemic preconditioning (RPc, in which a limb is made transiently ischaemic before the cardiac ischaemia) has a cardioprotective action that is mediated by relaxation of pericytes[12].

[1]Department of Translational Health Sciences, Bristol Medical School, University of Bristol, Bristol, UK. [2]Department of Neuroscience, Physiology and Pharmacology, University College London, London, UK. [3]These authors jointly supervised this work: Svetlana Mastitskaya, David Attwell. ✉ e-mail: svetlana.mastitskaya@bristol.ac.uk; d.attwell@ucl.ac.uk

These results suggest that, if the mechanism by which RPc relaxes coronary pericytes could be defined, it might offer a pharmacological therapeutic approach superior to the use of adenosine, which lowers blood pressure and has other side effects. We have previously discovered that the cardioprotective action of RPc is mediated by a soluble factor released by activity in the motor fibres of the posterior gastric branch of the vagus nerve[13,14], which was suggested[15] to be glucagon-like peptide 1 (GLP-1).

We now demonstrate, by examining cardiac ischaemia in vivo and imaging pericyte-mediated coronary capillary constriction in the pressurised vasculature of excised ventricle, that the protective effect of RPc on coronary blood flow involves release of GLP-1, which relaxes coronary pericytes by activating $K_{ATP}$ channels in these cells. The magnitude of the resulting blood flow increase and cardiac protection can be modulated by agents that alter trafficking of $K_{ATP}$ channels to the surface membrane, including AMP kinase, NO and acetylcholine.

## Results

### GLP-1 receptors are on pericytes on coronary capillaries

The fluorescent GLP-1R-binding compound[16] LUXendin555 bound to pericytes (and also some endothelial cells) in the mouse coronary vasculature, suggesting the presence of GLP-1 receptors on pericytes (Fig. 1a). Similarly, an antibody against GLP-1R also labelled mouse and cultured human coronary pericytes (Fig. 1b, c).

### No-reflow after coronary ischaemia reflects capillary constriction

Occluding the rat left anterior descending coronary artery (LAD) for 45 min and then allowing 15 min reperfusion induced a long-lasting 40% decrease of coronary blood volume in the part of the left ventricle supplied by the LAD, as compared with sham-operated rats ($p = 2.2 \times 10^{-7}$, Fig. 2a, c). This correlated with an increase in the percentage of capillaries blocked in the core of the area at risk (1–2 mm deep in the left ventricular wall: Fig. 2b, f) from $11.6 \pm 1.5\%$ ($n = 3$ hearts) to $73.9 \pm 9.0\%$ ($n = 6$, $p = 0.0001$, one-way ANOVA with Tukey's test).

We have previously shown that this capillary block and reduction of perfused blood volume are produced by pericyte-mediated constriction[8]. As previously, the diameter of capillaries at pericyte somata was greatly reduced (by ~45%) after ischaemia and reperfusion ($p = 0.017$ by one-way ANOVA in 48 capillaries in 3 sham-operated hearts and 48 capillaries in 3 hearts exposed to ischaemia-reperfusion, Fig. 2g).

### RPc reduces no-reflow via GLP-1R-mediated pericyte relaxation

Remote ischaemic preconditioning prevented the ischaemia-evoked pericyte-mediated capillary constriction ($p = 0.009$, Fig. 2g), and (presumably as a result) reduced capillary blockage from $73.9 \pm 9.0\%$ to $30.7 \pm 5.0\%$ of capillaries blocked ($p = 0.0016$, in 5 hearts) compared to coronary ischaemia/reperfusion alone (Fig. 2d–f). RPc also restored the post-ischaemic perfused blood volume ($p = 1.7 \times 10^{-7}$ compared with ischaemia/reperfusion alone) to a level similar to that seen for sham-operated mice (Fig. 2c).

Applying the GLP-1R blocker Exendin(9–39) (Ex9) intravenously, before the ischaemia, prevented RPc-mediated reduction of capillary block (resulting in $69.9 \pm 6.1\%$ capillary block, $n = 7$ hearts, $p = 0.003$ compared with RPc+ischaemia/reperfusion, Fig. 2f), and led to an ischaemia-evoked fall of perfusion volume similar to that seen with ischaemia/reperfusion alone ($p = 1.7 \times 10^{-5}$ compared to RPc+Ex9 group, Fig. 2c). If applied with ischaemia/reperfusion in the absence of RPc, Ex9 had no effect (Fig. 2c, f), consistent with it acting solely by preventing the effects of RPc.

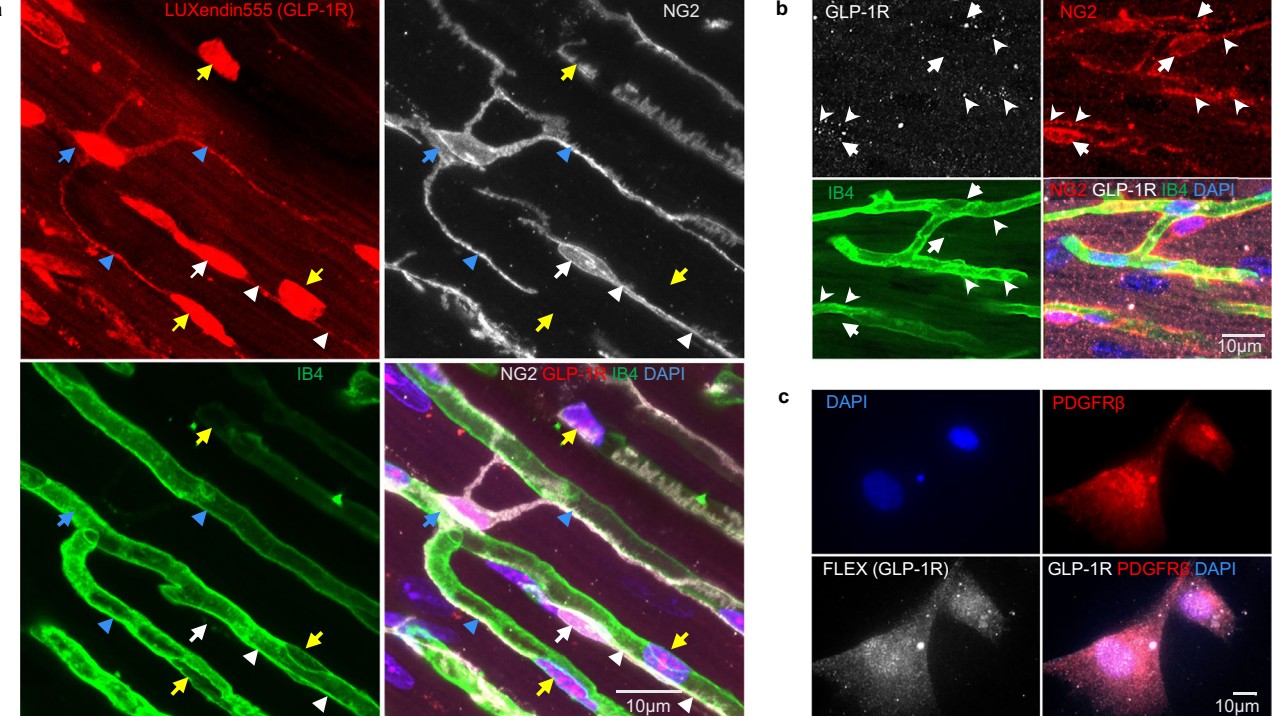

**Fig. 1 | Cardiac pericytes express GLP-1 receptors. a** Labelling of mouse coronary microvasculature with LUXendin555 (red) which labels GLP-1R. NG2 antibody labelling (white) shows that LUXendin-labelled cells are pericytes (white arrow and arrowheads show somata and processes respectively), telocytes (blue arrow/arrowheads, which may function similarly to pericytes but have processes contacting more than one capillary) or endothelial cells (yellow arrows). IB4 labelling (green) shows the basement membrane around endothelial cells and pericytes. The merge of these labels (bottom right) also includes nuclear labelling with DAPI (blue). **b** Labelling of mouse coronary microvasculature with anti-GLP-1R antibody (white), NG2 (red), IB4 (green) and DAPI (blue). White arrows and arrowheads show pericyte somata and processes, respectively. **c** Labelling of cultured human cardiac pericytes with Fluorescein-Trp25-Exendin-4 (FLEX) probe binding to GLP-1R, along with PDGFRß (red) and nuclear (DAPI, blue) labelling.

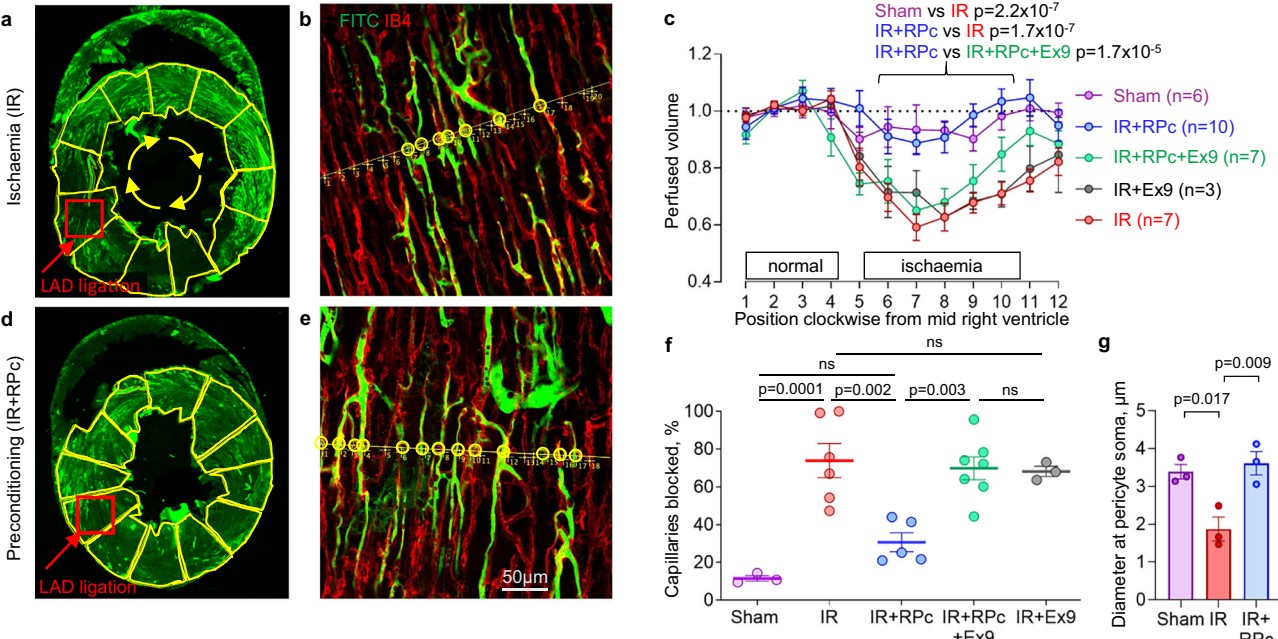

**Fig. 2 | RPc prevents ischaemia/reperfusion-induced capillary constriction and no-reflow via GLP-1R activation. a** Low-power view of a rat heart cross-section after ischaemia/reperfusion (IR). **b** Zoomed-in image taken from part of the area outlined red in (**a**), with a line drawn across capillaries to calculate the ratio of blocked to total capillaries; perfused capillaries are circled. **c** Perfusion volume assessed as mean FITC-albumin fluorescence in yellow ROIs from (**a**), indexed clockwise (as seen from above) around the left ventricular wall and normalised to the average of the first 3 ROIs (one-way ANOVA with Šídák's test for ROIs 6–10).

**d**, **e** As in (**a**, **b**), but with RPc applied before ischaemia, restoring perfusion (**c**). **f** IR increases capillary block in the anterior left ventricular wall compared to Sham; RPc reduces this (IR+RPc), and the RPc effect is blocked by the selective GLP-1R antagonist Exendin(9–39) (Ex9); Ex9 alone (IR+Ex9) has no effect in ischaemia (one-way ANOVA with Tukey's test). **g** Capillary diameter at pericyte somata after Sham, IR, and IR+RPc (one-way ANOVA with Šídák's test). Data are mean ± SEM; hearts are the statistical unit. Source data are provided as a Source data file.

## GLP-1R-evoked pericyte relaxation is via activation of $K_{ATP}$ channels

To investigate the mechanism by which GLP-1 receptor activation protects against the ischaemia/reperfusion-induced pericyte contraction[8,12] that blocks capillaries and reduces coronary blood flow, we used an ex vivo preparation. This comprised mouse free right ventricular wall, which we perfused through the vasculature by cannulating the right coronary artery (see 'Methods' and Fig. 3a) in addition to superfusing it with extracellular fluid. Capillary diameters were measured at pericyte somata (Fig. 3b). Oxygen and glucose deprivation (OGD) for 25 min evoked pericyte contraction and a constriction of capillaries, with a decrease of capillary diameter by 15.9 ± 1.7% ($p = 0.0002$, at $n = 21$ pericytes in tissue from 6 hearts, compared to the initial diameter, paired t-test, Fig. 3c). In continuous OGD conditions, the capillary diameter continued to gradually decline to 23.0 ± 2.6% less than the initial value (Fig. 3c).

Activating GLP-1R with Exendin-4 (Ex4, 100 nM) in the continued presence of OGD, starting after 25 min of OGD, relaxed pericytes within 25 min and dilated capillaries back to a diameter that was similar to the baseline value before ischaemia/reperfusion (Fig. 3c, constricted by 5.4 ± 1.0% at $n = 25$ pericytes in 7 hearts, compared to 23.0 ± 2.6% in continuous OGD without Ex4 at $n = 21$ pericytes in 6 hearts; $p = 6.1 \times 10^{-6}$, hearts are the statistical unit, one-way ANOVA with Šídák's test on final-minute values). However, in the presence of the ATP-sensitive K+ channel blocker glibenclamide (20 μM), the GLP-1R-mediated pericyte relaxation was abolished, and capillaries remained constricted by 19.0 ± 1.4% of the initial diameter (at $n = 23$ pericytes in 6 hearts, $p = 1.2 \times 10^{-4}$ compared to Ex4 only, Fig. 3c). This is consistent with RPc-evoked cardiac protection being blocked by glibenclamide in vivo[17] and with GLP-1 activating $K_{ATP}$ channels in smooth muscle cells[18] and pancreatic beta cells[19]. Similarly, in the presence of the GLP-1R antagonist Exendin(9–39) (Ex9, 100 nM), Ex4

failed to relax cardiac pericytes (capillary diameters remained reduced by 16.2 ± 2.6% of the initial diameter at $n = 22$ pericytes in 5 hearts, $p = 5.1 \times 10^{-4}$ compared to Ex4 only, one-way ANOVA with Šídák's test on final-minute values, Fig. 3d).

ATP-sensitive K+ channels in vascular mural cells (smooth muscle cells around arterioles and pericytes around capillaries) have previously been shown to play a major role in regulating vascular tone[20]. Ventricular cardiac pericytes express[21] both Kir6.1 (a key component of ATP-sensitive K+ channels[20,22]) and SUR2 (an essential subunit of coronary pericyte $K_{ATP}$ channels[23]). To confirm the involvement of ATP-sensitive K+ channels in the regulation of capillary diameter by pericytes, pericytes were preconstricted for 20 min with endothelin-1 (ET1, 20 nM, applied via the capillary lumen). ET1 decreased the capillary diameter at pericyte somata by 29.6 ± 3.3% (at 7 pericytes in 2 hearts after 20 min, Fig. 3e). Applying the $K_{ATP}$ channel opener pinacidil[22] (100 μM, in the capillary lumen) after terminating ET1 application resulted in a relaxation of pericytes and an increase of capillary diameter back to the baseline level within 20 min (Fig. 3e). In contrast, in tissue from mice with Kir6.1 knocked out in pericytes (see 'Methods'), capillaries remained constricted in the presence of pinacidil (by 31.3 ± 2.9% of the initial diameter, at $n = 12$ pericytes in 4 hearts, compared to 7.5 ± 2.8% in WT, $p = 0.007$, t-test on final-minute values, Fig. 3e). Similarly, the reversal of OGD-induced capillary constriction by Ex4 (Fig. 3f) was inhibited when Kir6.1 was knocked out in pericytes, so that capillaries remained constricted after 50 min of OGD at 21.4 ± 1.1% of the initial diameter, at $n = 9$ pericytes in 3 hearts, $p = 1.7 \times 10^{-5}$ compared to WT, t-test on final-minute values (Fig. 3f). Consistent with these results in isolated ventricles suggesting a role for ATP-sensitive K+ channels in mediating the effects of GLP-1 during RPc, in mice with Kir6.1 knocked out in pericytes, RPc was not protective against ischaemia/reperfusion injury (Fig. 3g).

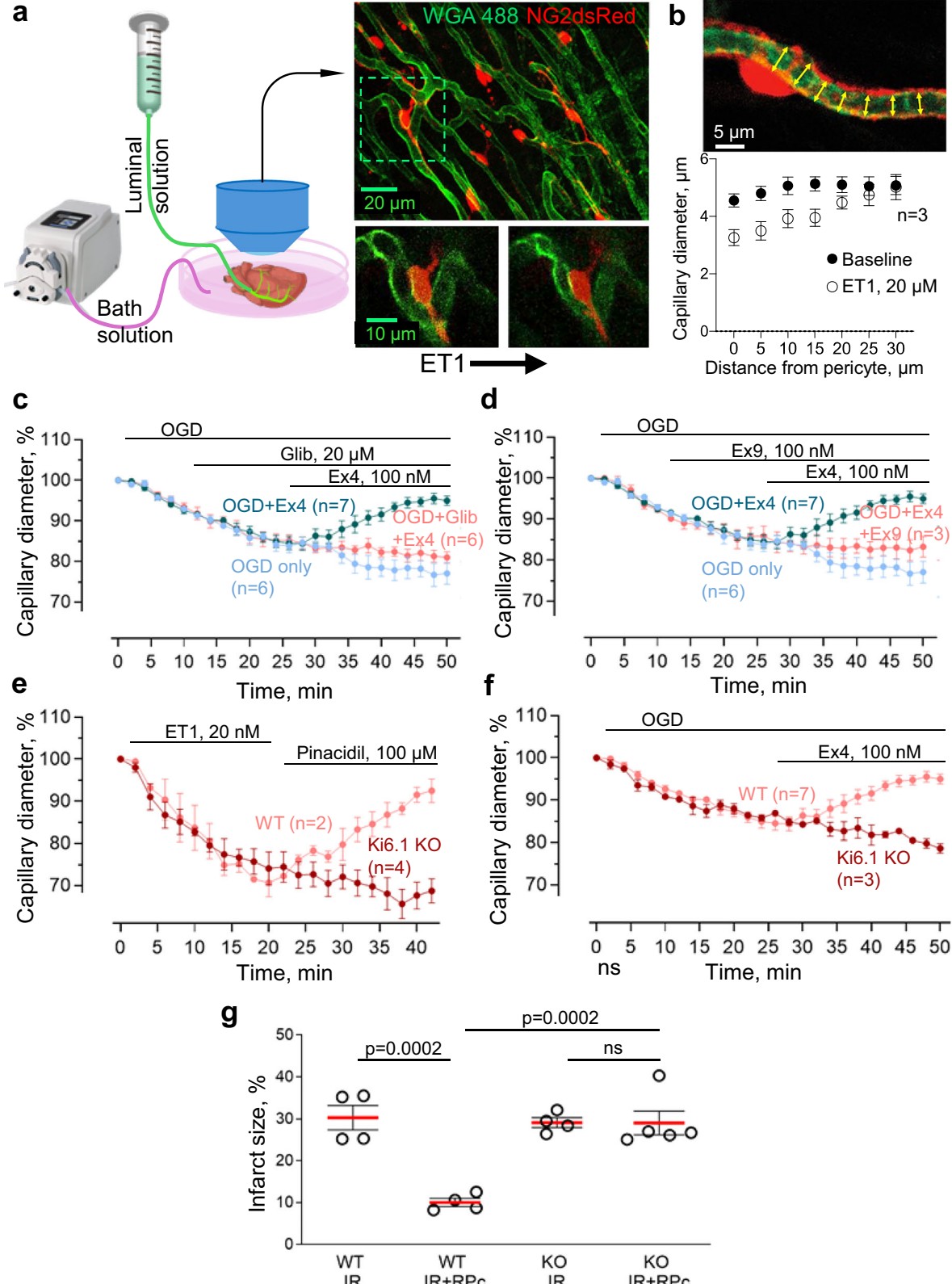

## Modulation of the GLP-1 - $K_{ATP}$ pathway by AMP kinase, NO and ACh

The ability of GLP-1 released by RPc to relax cardiac pericytes and increase coronary blood flow presumably depends on the activation of $K_{ATP}$ channels in the surface membrane of the pericytes, which will hyperpolarise the pericytes and decrease voltage-gated $Ca^{2+}$ entry. AMP-dependent protein kinase (AMPK) contributes to cell survival during cardiac ischaemia[24] and is known to promote activation of $K_{ATP}$ channels[25], at least in part by increasing trafficking of these channels to the surface membrane[26]. We found that applying compound C (CC, $5\,\mu M$), an AMPK blocker[27], inhibited the GLP-1R-mediated reversal of OGD-induced pericyte-mediated constriction of coronary capillaries ($p = 6.4 \times 10^{-4}$ at 15 pericytes in 4 hearts, hearts are the statistical unit, one-way ANOVA with Šídák's test on final-minute values, Fig. 4a). This

**Fig. 3 | GLP-1 receptors dilate capillaries by activating K$_{ATP}$ channels.**
**a** Schematic of the perfused mouse right ventricle: tissue pinned in chamber, luminally perfused via the right coronary artery and superfused. Capillary diameters at pericyte somata were measured during vasoactive agent application (e.g. endothelin-1, ET1). **b** Capillary diameter changes as a function of distance from pericyte somata in response to ET1 ($n$ = 3 hearts). **c, d** Effect of oxygen-glucose deprivation (OGD) on capillary diameter at pericytes is reversed by the GLP-1 receptor activator Exendin-4 (Ex4), but this rescue of diameter is negated by the K$_{ATP}$ channel blocker glibenclamide and in the presence of the GLP-1R antagonist Exendin(9–39) (Ex9). **e** ET1-induced constriction in wild-type (WT) tissue is reversed by the K$_{ATP}$ activator pinacidil, but this reversal is abolished by knockout of the K$_{ATP}$ component Kir6.1. **f** OGD-induced constriction in WT tissue was reversed by Ex4, but this rescue of diameter was prevented by KO of Kir6.1. **g** Myocardial infarct size evoked by 30 min ischaemia and 90 min reperfusion in vivo (IR) is reduced by RPc in WT, but not in Kir6.1 KO mice, consistent with a role for ATP-sensitive K$^+$ channels in mediating the effects of GLP-1 during RPc (one-way ANOVA with Tukey's test). Data are mean ± SEM; hearts are the statistical unit. Source data are provided as a Source data file.

is consistent with compound C inhibiting the cardiac protection provided by remote ischaemic conditioning, resulting in a larger infarct size[28].

The nitric oxide synthase (NOS) blocker L-NAME (100 μM) did not affect the constriction evoked by OGD alone, but blocked the capillary dilation evoked by activating GLP-1 receptors in OGD ($p$ = 0.0037 at 19 pericytes in 3 hearts, one-way ANOVA with Šídák's test on final-minute values, Fig. 4b). This is consistent with NOS block preventing the beneficial effect of RPc on no-reflow in vivo[29]. NO might be expected to dilate blood vessels directly by relaxing capillary pericytes or preventing them constricting[9], but this would be expected to occur even in the absence of GLP-1 receptor activation, predicting a constriction when L-NAME is applied, which is not seen in Fig. 4b. Instead, it is possible that the effect of NOS block reflects a decrease of trafficking of K$_{ATP}$ channels to the membrane because NO has been reported to activate AMPK[30,31]. Atropine (1 μM) also did not affect the constriction evoked by OGD alone, but blocked the capillary dilation evoked by activating GLP-1 receptors in OGD ($p$ = 0.0033 at 25 pericytes in 5 hearts, one-way ANOVA with Šídák's test on final-minute values, Fig. 4c), consistent with atropine blocking RPc-evoked cardiac protection after ischaemia[15]. Again, the lack of effect on the OGD-evoked constriction but suppression of the Ex4-evoked dilation might reflect acetylcholine promoting trafficking of K$_{ATP}$ channels to the membrane, because the type 3 muscarinic receptors that are cardioprotective in RPc[15] are reported to increase activation of AMPK[32,33].

To assess whether AMPK does indeed regulate K$_{ATP}$ channel trafficking, we used an antibody to Kir6.1, which forms an essential subunit of coronary pericyte K$_{ATP}$ channels[23] (Fig. 4d, e). In mouse hearts subjected to 40 min LAD ligation followed by 10 min reperfusion, the peak fluorescence intensity of Kir6.1 labelling shifted inside the cell in animals administered with the AMPK inhibitor compound C (CC, 10 mg/kg i.p.), consistent with AMPK mediating insertion of these channels into the surface membrane (Fig. 4f, g). This occurred both in ischaemia/reperfusion (1.11 ± 0.03 μm in the IR + CC group compared to 0.75 ± 0.03 μm in the IR control; $p$ = 0.001, one-way ANOVA with Šídák's test), and in the animals pre-treated with Exendin-4 (Ex4, 5 μg/kg i.p.; 1.13 ± 0.06 μm in the IR + CC + Ex4 group vs 0.67 ± 0.07 μm in the IR+Ex4 group; $p$ = 0.0004).

## Discussion

A signalling pathway consistent with the data presented above is schematised in Fig. 5. Ischaemia in vivo or OGD ex vivo inhibits Ca$^{2+}$ pumping out of pericytes, activating pericyte contraction and capillary constriction. Ischaemic preconditioning releases GLP-1 from the gut[13–15], which activates GLP-1 receptors on pericytes, raising the concentration of cAMP[34] or activating PI3 kinase in the pericytes[15,19,35] and thus activating K$^+$-ATP channels (provided AMPK activation[24] has inserted them into the surface membrane). K$^+$ efflux then hyperpolarises the pericyte membrane, inhibiting Ca$^{2+}$ influx through voltage-gated Ca$^{2+}$ channels and thus relaxing pericytes.

Our results provide a potential therapeutic approach for dealing with the fact that, following coronary ischaemia, lack of reflow in the microcirculation after an upstream artery is unblocked has a negative impact on clinical outcome. Here we have confirmed our earlier

discovery that no-reflow after coronary ischaemia reflects pericyte-mediated constriction of capillaries[8,12] (Fig. 2). This constriction will reduce blood flow by increasing the vascular resistance (both according to Poiseuille's law and because the viscosity of the blood increases at small vessel diameters due to interactions of the cells suspended in the blood with the capillary wall[36]). In addition, it is likely that, as in the brain, there is an extra suppressive effect on blood flow by neutrophils and other cells stalling in capillaries near pericyte-induced constrictions[37,38].

Remote ischaemic preconditioning (RPc) is an inter-organ protection phenomenon whereby ischaemia of an organ distant from the heart protects the myocardium against ischaemia/reperfusion injury. This involves activation of parasympathetic vagal outflow, and release from the gut[14] of GLP-1[15], which, via the systemic circulation, reaches the heart and conveys cardioprotection. Cardioprotective factors have also been suggested to be released from the spleen under vagal control[39], but GLP-1 is not produced in the spleen[40,41], and a role for the spleen is controversial[42]. In rats, activating GLP-1 receptors with Ex4 during reperfusion after coronary ischaemia reduces infarct size[43], and in humans giving GLP-1 after myocardial infarction or in heart failure improves outcome[44,45], although the mechanism by which GLP-1 conveys cardioprotection has been unclear[46]. Here, we demonstrate that GLP-1R-mediated cardioprotection is at least partly due to its relaxing effect on pericytes on coronary capillaries. This is analogous to GLP-1R-mediated relaxation of the coronary arteries[18] and aorta[34].

The long-lasting ischaemia-evoked constriction of capillaries by pericytes, and hence no-reflow, was reduced by RPc (Fig. 2g), which evokes GLP-1 release[15] and by the GLP-1 analogue Ex4, which is presumed to activate pericyte GLP-1 receptors and thereby activate ATP-sensitive K$^+$ channels (Fig. 3) as in pancreatic beta cells[19] (Fig. 5). K$^+$ efflux through these channels will hyperpolarise the pericyte. In turn, this will decrease activation of voltage-gated Ca$^{2+}$ channels, relaxing pericytes and thus dilating capillaries near the pericyte somata, where the majority of pericytes' circumferential processes are located[47]. Involvement of ATP-sensitive K$^+$ channels was shown both with pharmacological blockers and with pericyte-specific knockout of these channels. A similar restoration of capillary diameter towards normal size is seen on administering adenosine after coronary ischaemia[8], probably also due to activation of ATP-sensitive K$^+$ channels[48], and giving adenosine or the ATP-sensitive K$^+$ channel opener nicorandil improves outcome in human percutaneous coronary angioplasty by preventing no-reflow and thus improving ventricular function[49–51].

Consistent with our LUXendin555 localisation data (Fig. 1), we assume for simplicity that the GLP-1 receptors mediating RPc are at least in part on the pericytes themselves, in order for them to adjust capillary diameter by altering pericyte contractile tone. The dilation of coronary capillaries by Exendin-4 in OGD is consistent with this GLP-1R agonist increasing coronary blood flow in humans[52]. It has previously been suggested that GLP-1 receptors are present on cardiac myocytes[53], but deleting any such receptors did not abolish the protective effect of Exendin-4 on coronary ischaemia[53] (which we attribute to increased coronary blood flow: Fig. 3). It has also been suggested that GLP-1Rs are located on endothelial cells of the endocardium[54]. While our imaging data (which were on the outer third of the

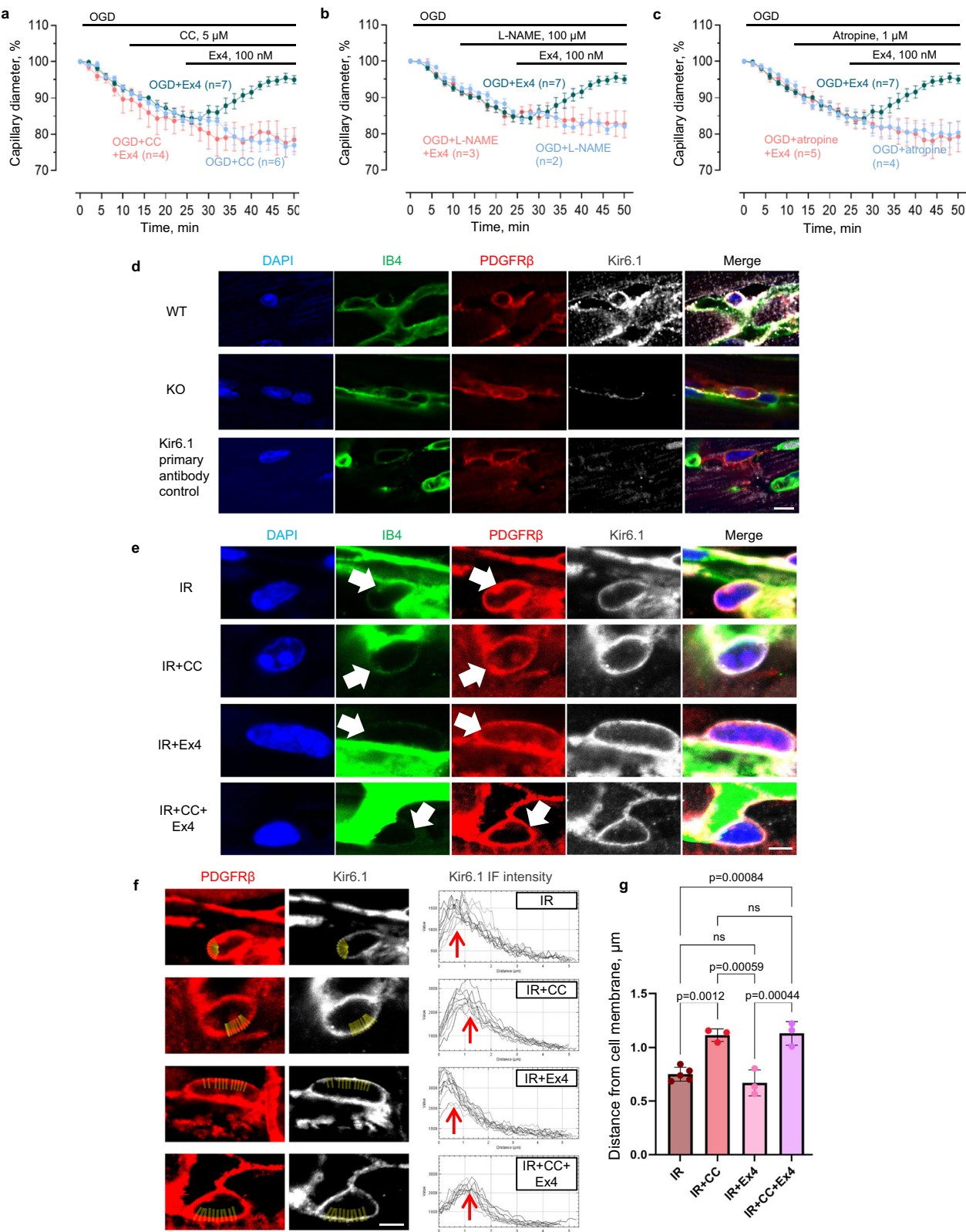

**f** PDGFRβ  Kir6.1  Kir6.1 IF intensity

**g**

ventricular myocardium where capillaries run more parallel to the section surface, making it easier to identify pericytes) suggested the presence of GLP-1 receptors on both pericytes and endothelial cells (Fig. 1), in any case gap junctional connections between endothelial cells and pericytes[55] could conceivably allow activation of GLP-1Rs and $K_{ATP}$ channels on endothelial cells to evoke hyperpolarisation of pericytes and thus to produce capillary dilation.

In order for activation of ATP-sensitive $K^+$ channels to hyperpolarise the cell and thus relax pericytes, these channels need to be present in the surface membrane. It is unclear whether these channels are in the surface membrane normally, or whether they are inserted into the membrane only in conditions of ischaemia when AMP kinase activation is increased[24]. Blocking AMPK was found to remove the capillary-dilating protective effect of GLP-1R activation in OGD

**Fig. 4 | GLP-1-evoked capillary dilation in ischaemia is modulated by AMP kinase, NO and cholinergic signalling. a** Exendin-4 reverses OGD-evoked mouse capillary constriction at pericytes; this is abolished by the AMPK blocker compound C (CC). **b** L-NAME does not affect OGD-evoked constriction, but abolishes GLP-1R-mediated dilation. **c** Atropine has no effect on the OGD-evoked constriction, but blocks the GLP-1R-mediated dilation in OGD. **d** Immunostaining of cardiac pericytes for Kir6.1 expression: cardiac tissue from wild type (WT), NG2-Cre$^{+/-}$/Kir6.1$^{flx/flx}$ (KO) mice and WT without Kir6.1 primary antibody. **e** Immunolabelled cardiac pericytes from four experimental groups of mice with LAD ligation: ischaemia/reperfusion only (IR), IR group pre-treated with CC (IR + CC), IR group pre-treated with Exendin-4 (IR+Ex4), and IR with both CC and Ex4 (IR + CC + Ex4). IB4 labelling extends all

around the pericytes (white arrows). **f** Examples of Kir6.1 immunofluorescence intensity profiles measured with ImageJ, using PDGFRβ immunostaining to define cell surface. 10 line ROIs per cell (yellow lines) were drawn from the membrane inward in the PDGFRβ channel, transferred to the Kir6.1 channel, and fluorescence intensity averaged per pericyte. Distance from membrane to peak Kir6.1 immunofluorescence was averaged per heart ($n = 40, 43, 30, 31$ pericytes in $N = 5, 3, 3$ and 3 hearts for IR, IR + CC, IR + Ex4 and IR + CC + Ex4). **g** Mean distance from membrane to peak Kir6.1 immunofluorescence intensity (also shown as red arrows in **f**). One-way ANOVA with Šídák's test. Data are mean ± SEM. Scale bar: 10 μm. Source data are provided as a Source data file.

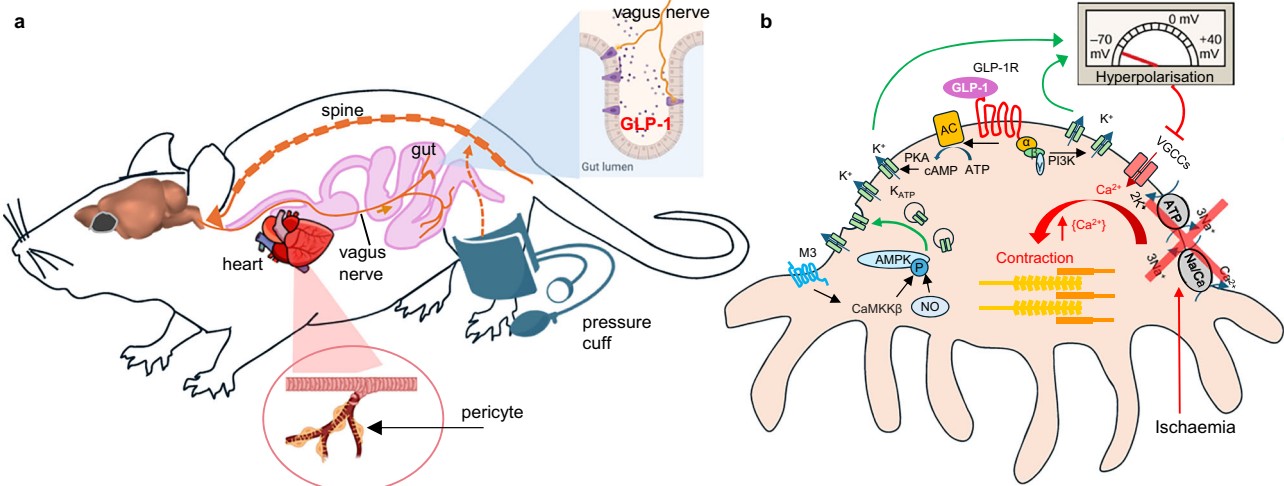

**Fig. 5 | Signalling pathways characterised in this paper. a** Schematic representation of brain-gut-heart neuro-humoral pathway in RPc cardioprotection. Hind limb ischaemia activates the RPc reflex via the spine, processed at the level of the brainstem. The neural stimulus is then relayed via the vagus to the gut, where GLP-1 is produced by enteroendocrine L-cells (inset: L-cells − purple; vagus − orange/brown). GLP-1 enters the systemic circulation with the blood/lymph and reaches the heart where it acts on the microvasculature to relax pericytes. **b** Molecular mechanisms downstream of GLP-1R activation on cardiac pericytes. Ischaemia

in vivo or OGD ex vivo depletes intracellular ATP and inhibits Ca$^{2+}$ pumping out of pericytes, activating pericyte contraction and capillary constriction. Ischaemic preconditioning releases GLP-1 from the gut, which activates GLP-1 receptors on pericytes, raising the concentration of cAMP or activating PI3 kinase, and thus activating K$^+$-ATP channels (provided AMPK activity has inserted them into the surface membrane). K$^+$ efflux then hyperpolarises the pericyte membrane, inhibiting Ca$^{2+}$ influx through voltage-gated Ca$^{2+}$ channels and thus relaxing pericytes. The activity of AMPK depends on muscarinic (M3) and nitric oxide mechanisms.

(Fig. 4a), probably because OGD-evoked AMPK activity[24] is needed to insert ATP-sensitive K$^+$ channels into the membrane[25,26], and block of AMPK is known to prevent the reduction of infarct size by RPc[28]. Consistent with the protective effect of AMPK, knockout or pharmacological block of the G-protein coupled receptor GPR39 was previously found to promote reflow and cardiac protection after ischaemia[56]. GPR39 inhibits cardiac AMPK[57], so knockout or block of GPR39 is expected to increase the surface membrane density of K$_{ATP}$ channels in pericytes. In accord with this, with GPR39 not functioning, the diameter of capillaries was larger near pericyte somata[56]. Similarly, both NO and ACh muscarinic receptors are known to activate AMPK[30–33] and although blockers of NO synthase or of muscarinic receptors did not affect the capillary constriction induced by oxygen-glucose deprivation (Fig. 4b, c), they inhibited the capillary dilation evoked by GLP-1 receptor activation, consistent with these blockers preventing cardioprotection by RPc[15,29]. Our observation that the muscarinic blocker atropine inhibited the effect of GLP-1 receptor activation (Fig. 4c), in an isolated piece of ventricle, implies that there is tonic local release of ACh in the isolated heart, possibly via a non-neuronal cholinergic system in the ischaemic heart[58,59]. Consequently, damage-promoting effects of atropine in the whole animal[15] may reflect actions intrinsic to the heart as well as suppression of vagally-induced GLP-1 release from the gut.

The present study does not advocate for the use of remote ischaemic preconditioning as a cardioprotective strategy. Large

multicenter trials (including ERICCA and RIPHeart) showed no reduction in major adverse cardiac events with RPc[60]. However, most patients in these trials received the anaesthetic propofol, which suppresses autonomic neurotransmission[61]. The age-related decline in vagal tone and cardiometabolic impairment further limits efficacy. Given that the vagal pathway is crucial in mediating cardioprotection, it is not surprising that the studies performed under propofol anaesthesia resulted in neutral outcomes. In contrast, large-scale GLP-1R agonist outcome trials consistently show reduced risk of major adverse cardiovascular events, independently of comorbidities or weight loss achieved[62] suggesting that targeting downstream effector pathways of neural mechanisms of cardioprotection using GLP-1R agonists is a promising therapeutic strategy.

The increase in coronary blood flow that we suggest here is produced by GLP-1 and has similarities to the increase in cerebral blood flow suggested to result from GLP-1 release during remote ischaemic preconditioning evoked before stroke[63]. Nevertheless, GLP-1 analogues, while improving blood flow in pathology-affected tissues, may also have side effects, for example, on the pancreas and gastrointestinal tract. Indeed, even in the heart, GLP-1R activation has been suggested to have actions other than opposing the effects of ischaemia-evoked blood flow reduction[64]. An increasing number of GLP-1 analogues are now being used in clinical practice, for disorders ranging from type 2 diabetes to obesity management and Alzheimer's

disease. It would be desirable to develop a version of these drugs that specifically targets pericyte-mediated reduction of coronary blood flow, for use in cases of coronary ischaemia.

## Methods

All experiments were performed in accordance with the European Commission Directive 2010/63/EU (European Convention for the Protection of Vertebrate Animals used for Experimental and Other Scientific Purposes) and the UK Home Office Scientific Procedures Act (1986), with project approval from the University College London Animal Welfare and Ethical Review Body (Bloomsbury Campus).

Animals were group-housed and kept on a 12 h light/dark cycle at 24 °C ambient temperature and 55-65% relative humidity (mice) or at 22 °C ambient temperature and 45–65% relative humidity (rats). Normal rodent chow (Teklad 2018 or 7912, Envigo) and tap water were available ad libitum.

The experiments involving human cells comply with the guidelines of the Declaration of Helsinki. Discarded right ventricular tissue from congenital heart defect surgery (pulmonary valve repair) was obtained with patients' parents or legal guardians informed consent (ethical approval 15/LO/1064 from the North Somerset and South Bristol Research Ethics Committee).

### Animal preparation

Adult male Sprague-Dawley rats weighing 220–250 g or transgenic mice (see below) were anaesthetised with pentobarbital sodium (induction 60 mg/kg i.p. for rats, 90 mg/kg i.p. for mice; maintenance 10–15 mg/kg/h i.v.). The right carotid artery and jugular vein were cannulated for blood pressure (BP) measurement and drug administration, respectively. Stable BP and heart rate were maintained, and level of anaesthesia was monitored by paw pinch response. The trachea was cannulated, and animals were mechanically ventilated with room air using a positive pressure ventilator (Harvard rodent ventilator, tidal volume of 1 ml/100 g of body weight, ventilator frequency ~60 strokes min$^{-1}$ for rats, ~150 strokes min$^{-1}$ for mice). Arterial BP and ECG were recorded using a Power1401 interface and *Spike2* software (Cambridge Electronic Design), and body temperature was maintained at 37.0 ± 0.5 °C. The heart was exposed via a left thoracotomy, and a 4-0 (rats) or 8-0 (mice) Prolene suture was passed around the left anterior descending (LAD) coronary artery to induce a temporary occlusion. Rats were subjected to 45 min of LAD ligation followed by 15 min of reperfusion to study no-reflow phenomena. To determine the effects of Exendin-4 (a GLP-1R agonist) and Compound C (an AMPK inhibitor, 6-[4-(2-Piperidin-1-yl-ethoxy)-phenyl]-3-pyridin-4-yl-pyyr-azolo[1,5-a] pyrimidine) on the localisation of K$_{ATP}$ channels in ischaemia, mice were subjected to 40 min of LAD ligation followed by 10 min of reperfusion, with Compound C (Dorsomorphin 2HCl, APExBIO; 10 mg/kg, i.p., administered 1 h before LAD ligation) and/or Exendin-4 (Tocris; 5 μg/kg i.p., given 30 min before LAD ligation). To determine the extent of myocardial ischaemia/reperfusion damage, mice were subjected to 30 min of LAD ligation followed by 90 min reperfusion. Successful LAD occlusion was confirmed by paling of the myocardial tissue distal to the suture, elevation of the ST-segment in the ECG, and an immediate 15–30 mmHg fall in the BP. To establish remote ischaemic preconditioning, blood supply to the lower limbs was interrupted for 15 min by placing clamps on both femoral arteries at the proximal level ~1 cm below the inguinal ligament 25 min before the onset of coronary ischaemia. The sham-RPc procedure involved dissection of both femoral arteries without occlusion. GLP-1R antagonist Exendin(9–39) (Tocris, Ex(9–39), 50 μg/kg, i.v.) or vehicle control was administered 40 min before the onset of coronary ischaemia. Control (sham-operated) animals underwent the same procedures, except that after the suture was passed around the LAD coronary artery, it was not drawn tight to occlude the vessel. Ischaemia and sham animals were alternately interleaved.

### Infarct size measurement

In mice, at the end of reperfusion period, the LAD was re-ligated, and 5% Evans Blue dye solution (100 μl) was infused via jugular vein to determine area at risk. The animal was then given an anaesthetic overdose (pentobarbital sodium, 200 mg/kg, i.v.), the heart was excised, briefly frozen and sectioned into 5–6 transverse slices from apex to base. The area at risk was demarcated by the absence of Evans Blue staining. Heart slices were then incubated with 1% 2,3,5-triphe-nyltetrazolium chloride (TTC) in Tris buffer (pH 7.4) for 15 min at 37 °C and fixed in 4% formalin for 24 h. Viable myocardium is stained red by TTC, whereas necrotic myocardium appears white. The area at risk and the necrotic area were determined by computerised planimetry, normalised to the weight of each slice, with the degree of necrosis (i.e. infarct size) expressed as the percentage of area at risk, as described[13,14].

### Animal perfusion and tissue processing for imaging

After the ischaemia/reperfusion procedure, rats were given an overdose of pentobarbital sodium (200 mg/kg, i.v.) and transcardially perfused with 200 ml saline followed by 200 ml 4% paraformaldehyde (PFA) for fixation and then by a solution of 5% gelatine (Sigma-Aldrich, G2625) containing 0.25% FITC-albumin conjugate (Sigma-Aldrich, A9771) for visualisation of the perfused coronary microvasculature. The initial perfusion with calcium-free saline ensures that the heart is stopped in diastole, which is also evident from the large volume of left ventricle visible at cross-section. The hearts were then fixed overnight in 4% PFA, and 150 μm transverse sections were obtained for immunofluorescence staining. The labelling was done using anti-NG2 antibodies (Merck Millipore, AB5320, 1:200) for pericytes and isolectin B4-Alexa Fluor 647 (Molecular Probes, I32450, 1:200) for the capillary basement membrane.

Mice used in Kir6.1 membrane localisation study were given an anaesthetic overdose (pentobarbital sodium, 200 mg/kg, i.p.) at the end of 10 min reperfusion and transcardially perfused with 20 ml saline, followed by 5 ml 4% PFA for fixation, the hearts were fixed overnight in 4% PFA and sliced into 150 μm transverse sections. The labelling was done using anti-PDGFRß antibodies (R&D Systems, AF1042, 1:50) for pericytes, isolectin GS-IB4-biotinylated (Life Technologies, 121414, 1:200) for capillary basement membrane, and anti-Kir6.1 (Alomone Labs, APC-105, 1:100) to localise the K$_{ATP}$ channels, following heat-induced epitope retrieval (using 10 mM sodium citrate buffer at pH 6.0 for 20 min at 80 °C).

To visualise GLP-1Rs in mouse pericytes, wild-type mice were injected with Luxendin555 probe subcutaneously[65] (100 pmol/g body weight) and 2 h after Luxendin555 administration, given an overdose of pentobarbital sodium (200 mg/kg, i.v.) and perfused-fixed (as described above). Tissue was processed as described above; the pericytes and blood vessels were labelled using anti-NG2 antibodies (Abcam, ab275024, 1:200) and isolectin B4-Alexa Fluor 488 (Molecular Probes, I21411, 1:200) respectively (Fig. 1a). In a separate set of experiments, heart tissue from NG2-dsRed mice was processed as described above and stained with anti-GLP-1R antibody (Abcam, ab218532, 1:200) following heat-induced epitope retrieval as above.

To visualise GLP-1Rs in human pericytes, pericytes isolated from cardiac surgery waste tissue (cell isolation methods and characterisation as described previously[66]) were cultured to semi-confluency, fixed with 4% PFA for 20 min and labelled with the anti-human PDGFRβ antibody (Santa Cruz, sc-374573, 1:50) and Fluorescent Exendin-4 (Fluorescein-Trp25-Exendin-4, FLEX; Eurogentec, AS-63899, 100 nM).

### Imaging vessels after in vivo ischaemia experiments

Images of the left ventricular capillary bed in rats were acquired using laser scanning confocal microscopy (Zeiss LSM 700) and analysed using FIJI software (ImageJ 1.53c, NIH). To quantify blood volume across the left ventricular myocardium, low-power z-stack images were taken of an

entire transverse section of each heart (using a 1× air objective), maximum intensity projected. 12 regions of interest (ROI) were drawn clockwise around the left ventricle (when looked at from above) from the mid-point of the septum[8] (as in Fig. 2a, d), and the mean intensity of FITC-albumin signal was recorded for each ROI and normalised to the highest intensity measured in any ROI. These data were averaged over hearts and renormalised so that the mean value in positions 1–3 of Fig. 2c was 1. This signal is assumed to be proportional to the volume of blood perfusing the myocardium. To compare perfusion in the ischaemic zone, we used the plotted values from ROIs 6 to 10.

For quantification of the percentage of capillaries that were perfused, three randomly selected regions of the outer third of the myocardium of the anterior left ventricular wall were imaged (with the capillaries running parallel to the surface of the tissue slice), as this region of myocardium consistently included the ischaemic area (which showed visible pallor and oedema of the myocardium). A line perpendicular to the capillaries was drawn in the central area of the image ($\pm 50 \, \mu m$), the numbers of the blocked and perfused capillaries were calculated, and the percentage of the blocked capillaries was averaged per heart. The person quantifying the images was blinded to the condition that the heart was exposed to. Blockages of flow in the ischaemic area at risk were identified by abrupt terminations in FITC-albumin signal (Fig. 2b, e).

### Transgenic mice

Some experiments used NG2-dsRed mice[67], kindly provided by Akiko Nishiyama (University of Connecticut) via Dirk Dietrich (University of Cologne), in which pericytes fluoresce red, facilitating their identification. Pericyte-specific Kir6.1 knockout mice were generated by crossing tamoxifen-inducible NG2-Cre$^{ERT2}$ knock-in mice[68] (kindly donated by Frank Kirchhoff, University of Saarland) and floxed (flx) Kir6.1 mice (kindly donated by Andrew Tinker, QMUL) to allow deletion of Kir6.1 in NG2-expressing pericytes after oral gavage of tamoxifen. Tamoxifen dissolved in corn oil was given at 100 mg/kg body weight by oral gavage once per day for 3 consecutive days to adult >P21 mice. Experiments were performed from 2 weeks after tamoxifen administration.

Genotypes were identified by polymerase chain reaction on genomic DNA from ear snips of NG2-cre$^{+/-}$/Kir6.1$^{flx/flx}$, and littermate control animals NG2-cre$^{+/-}$/Kir6.1$^{-/-}$ using the following primers: for NG2-Cre, WT sense AGAGATCCTGTCCACAGAGTTCT and antisense GCTGGAGCTGACAGCGGGTG and generic Cre, sense CACCCTGT-TACGTATAGCCG and antisense GAGTCATCCTTAGCGCCGTA (NG2 WT band 557 bp, Cre band 330 bp); for Kir6.1, sense GAGTCTTAACTCA GTTCTGGAGGACCAACA and antisense AGCGAAGAAAACTGCTTCC TGTTCATTAAG (Kir6.1 WT band 474 bp, flx band 600 bp).

### Live ex vivo tissue imaging

Adult mice (P100-P120) of both sexes were used in the study. The heart of a mouse freshly euthanised with isoflurane overdose was surgically removed and placed into ice-cold Tyrode's solution containing the following components (mM): 142.5 NaCl, 2.5 KCl, 0.5 MgCl$_2$, 0.33 NaH$_2$PO$_4$, 5 glucose, 2 Na lactate, 0.1 Na pyruvate, 10 Hepes and 1.8 CaCl$_2$ (pH set to 7.4 with NaOH). After being cleaned of connective tissues, the heart was transferred into a Sylgard-coated prechilled (4 °C) chamber containing Tyrode's solution and pinned down onto the Sylgard layer of the chamber. The right ventricular wall was dissected, and the right coronary artery exposed and cannulated using a fire-polished bent glass pipette prefilled with Tyrode's solution. The right ventricular wall was used for imaging as it was thinner than the left ventricular wall, which made it easier to flatten under the objective and ensured full immersion of the tissue in the bath superfusion solution. The pipette used to cannulate the artery was then connected to a fluid-filled line, and intralumenal pressure was created by elevating a cylinder connected to the line with Tyrode's solution to a height

equivalent to 80 mmHg (109 cmH$_2$O) (Fig. 3a). The vascular lumen perfusate was Tyrode's solution (composition as above). The bath superfusate contained a physiological saline solution (PSS; containing (mM) 114.5 NaCl, 2.5 KCl, 1.2 MgSO$_4$, 1.2 NaH$_2$PO$_4$, 24 NaHCO$_3$, 10 glucose, 1 Na lactate and 1.8 CaCl$_2$) was maintained at 32–34 °C with a heater and temperature controller and bubbled with 5% CO$_2$, 20% O$_2$ and 75% N$_2$ to maintain a pH of 7.4. The bath solution was perfused at ~5 ml/min. Experimental measurements were started after ~30–60 min stabilisation. To visualise the vessels, Alexa Fluor 488-conjugated WGA (Thermo Fisher Scientific, W11261; 20 µg/mL, 30 min) was included in the luminal buffer during the stabilisation period. For imaging of the tissue from animals not expressing dsRed in pericytes, an additional step of 20 min incubation with isolectin B4-Alexa Fluor 594 (Molecular Probes, I21413, 10 µg/ml in superfusate solution) was employed. To block cardiac myocyte contraction, a modulator of cardiac myosin, Mavacamten[69] (MYK-461, 10 µM, MedChemExpress), was added to the luminal solution.

Two-photon excitation used a Newport-Spectra Physics Mai Tai Ti:Sapphire Laser pulsing at 80 MHz, and a Zeiss LSM780 (Oberkochen, Germany) microscope with a 20× water immersion objective (NA 1.0). Fluorescence was excited using 920 nm wavelength for DsRed, and 820 nm for Alexa Fluor 488. Mean laser power under the objective was <35 mW. Z-stack images were acquired every 2 min. Baseline for each condition was 5 min. The ex vivo model of tissue ischaemia was induced by oxygen/glucose deprivation (OGD)—the tissue was exposed to perfusate solution saturated with 95% N$_2$/5% CO$_2$ gas mixture containing 10 mM sucrose instead of glucose. The period of OGD lasted for 25 min, after which GLP-1R agonist was added to the perfusate solution, and OGD continued for another 25 min. Exendin-4 (Ex4; 100 nM, Tocris) was used as a GLP-1R agonist. N(gamma)-nitro-L-arginine methyl ester (L-NAME; 100 µM, Sigma) was used to inhibit nitric oxide synthesis. Atropine sulphate (1 µM, Sigma) was used to block muscarinic receptors. Compound C was used to inhibit AMPK (5 µM, APExBIO). K$_{ATP}$ channels were activated with pinacidil (100 µM, AOBIOUS) and blocked with glibenclamide (20 µM, Santa Cruz). Endothelin-1 (20 nM) was applied either through the vascular lumen or in the superfusate, and the constriction evoked did not differ significantly (22.3 ± 7.3% constriction when ET1 was applied in the lumen solution compared to 24 ± 5.4% constriction when ET1 was applied in the bath solution). For mechanistic experiments, ET1 and pinacidil were applied via the capillary lumen, while all other drugs were applied in the superfusion solution.

### Image analysis after live ex vivo tissue imaging and immunofluorescence studies

Pericytes were identified by their morphology (spatially isolated cells located outside capillaries) as confirmed by IB4 labelling, or by antibody labelling for their characteristic markers NG2 or PDGFRβ, or by expression of dsRed under the NG2 promoter.

Capillary diameters and Kir6.1 immunofluorescence intensity profiles were analysed using FIJI software (ImageJ 1.53c, NIH) by a person blinded to the experimental condition. Vessel diameter was defined using a line drawn across the vessel as the width of the intra-luminal dye fluorescence (WGA-Alexa Fluor 488) at pericyte somata. To assess the Kir6.1 immunofluorescence intensity profiles, PDGFRβ immunostaining was used as a reference point to localise the cell surface. Between 3 and 5 zoomed-in images per heart were taken to achieve a total number of 5-10 pericytes per heart, and all imaged pericytes were analysed. The selection of the membrane area was based on clearly visible PDGFRβ signal, on the outer side of the capillary. 10 regions of interest (ROI) per pericyte were drawn as lines in the PDGFRβ channel, starting from the cell surface and directed towards the centre of the cell. The ROIs were then transferred to the Kir6.1 channel and the fluorescence intensity profile of each line was measured and averaged over all the lines per pericyte. The values of

the resulting distance from the cell membrane to the peak Kir6.1 immunofluorescence intensity were averaged for each heart and then used for the analysis (Fig. 4).

## Statistics and reproducibility

Statistical analyses were performed using GraphPad Prism 9 software. Data normality was assessed with Shapiro-Wilk tests. Comparisons of normally distributed data were made using two-tailed Student's t-tests. Equality of variance was assessed with an F test, and Welch's t-test was used when variances were unequal. *P* values were obtained using one-way ANOVA or t- tests and adjusted for multiple comparisons with the Šidák or Tukey method, as appropriate. Corrected *p* values < 0.05 were considered statistically significant. Numbers of animals are denoted as *N*, and numbers of pericytes or capillaries as *n*. In all statistical tests, animals/hearts (*N*) were the statistical unit. Data are presented as mean ± SEM.

Where representative examples are shown from immunostaining images, the staining was repeated independently at least twice with similar results. Specifically, for Fig. 1a, the LUXendin probe was administered and subsequently visualised in two mice. For Fig. 1b, hearts from two mice were stained with anti-GLP-1R antibodies. For Fig. 1c, cultured human cardiac pericytes from two different patients were labelled with Fluorescein-Trp25-Exendin-4 (FLEX) to visualise GLP-1Rs. For Fig. 2b, e, three areas from 3 to 5 hearts per group were imaged on different experimental days. For Fig. 4d, three hearts per group (WT and KO) were stained with anti-Kir6.1 antibody. For Fig. 4e, 3–5 hearts per group were imaged and analysed.

## Reporting summary

Further information on research design is available in the Nature Portfolio Reporting Summary linked to this article.

# Data availability

The data that support the findings of this study are available within the article or from the corresponding author upon request. The source data underlying Figs. 2–4 are provided as a Source data file. Source data are provided with this paper.

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

## Acknowledgements

The study was supported by the British Heart Foundation Intermediate Basic Science Research Fellowship (FS/IBSRF/21/25060) to S.M., and an ERC Advanced Investigator Award (740427, BrainEnergy), Wellcome

Trust Senior Investigator Award (099222/Z/12/Z), a Rosetrees Trust grant (M153-F2), and a BHF/UK-DRI Centre for Vascular Dementia Research grant to D.A. The authors thank Frank Kirchhoff for NG2-Cre mice, Akiko Nishiyama and Dirk Dietrich for NG2-dsRed mice, Andrew Tinker for Kir6.1$^{flx/flx}$ mice, Elisa Avolio for providing human pericyte culture, David Hodson and Johannes Broichhagen for providing Luxendin555 probe, Thomas Kampourakis for suggesting the use of Mavacamten, Stuart Martin for genotyping, and Wolfgang Langhans, Alice Adriaenssens and Sergey Kandabarau for advice during this work.

## Author contributions

S.M. and D.A. designed experiments. S.M. performed experiments and analysed the data. F.S.S.F. performed immunostaining for the study of perfusion volume and capillary blockages (Fig. 2), L.E.E. performed experiments and immunostaining for the Kir6.1 localisation study (Fig. 4d, e). S.M. and D.A. wrote the manuscript.

## Competing interests

The authors declare no competing interests.
