## [Transparent Peer Review file · Nature Communications]

GLP-1 activates K_{ATP} channels in coronary pericytes as the effector of brain-gut-heart signalling mediating cardioprotection

Corresponding Author: Dr Svetlana Mastitskaya

Version 1:

Reviewer comments:

Reviewer #1

(Remarks to the Author)

This is an interesting and well-written manuscript providing evidence in support of the idea that remote preconditioning (RPC) protects the heart after ischemia via GLP-1 activation of K_{ATP} channels on pericytes, leading to capillary dilation and protection against the no-reflow phenomenon. The data appear to be of high quality and, at face value, provide compelling support. I have a few comments, some relatively minor.

1. Many channels are indirectly regulated by ATP. To be consistent with the literature, it would be better to use the generally accepted term: ATP-sensitive potassium channels, rather than ATP-gated.
2. Figs. 2b and d, do not look like blowups of a and c. Are the images inverted somehow? How was it decided where to draw the lines in Fig 2 b and d? Moving them up or down could give very different results in 2f.
3. Antibodies against K_{ATP} channel subunits have been notoriously unspecific. The authors could carry out control experiments on Kir6.1 knockout tissue to confirm the identified channels in Fig. 4.
4. How strong is the evidence for the gut/CNS part of the proposed model in Fig. 5? GLP1 is synthesized in L-cells, but also in alpha cells. Could it be made elsewhere, and are the RPC effects lost with spinal lesion, for example?

Reviewer #2

(Remarks to the Author)

In the present study Mastitskaya and colleagues studied whether coronary microvascular pericytes participate in the previously reported protective vasodilating effects of "endogenous" glucagon-like peptide (GLP-1) following remote ischemic preconditioning. In analogy to these previous studies from Basalay MV, et al. ("Glucagon-like peptide-1 (GLP-1) mediates cardioprotection by remote ischemic conditioning. *Cardiovasc Res.* 2016;112:669-676. doi: 10.1093/cvr/cvw216") here the authors used a rat model of myocardial infarction (coronary artery occlusion followed by reperfusion). In such anesthetized rats, remote ischemic preconditioning (RPC) was induced by 15 min occlusion of femoral arteries applied 25 min prior to the myocardial ischemia. The present studies reproduce the effects observed by Basalay et al. showing that RPC attenuates the ischemia-induced reduction of coronary flow, and that this cardioprotective vasodilating effect is attenuated following blockade of GLP-1 receptors (GLP-1R) by Exendin(9-39) (Ex9, administered through systemic i.v. injection before LAD ligation). The novel aspect of the 1st part of the present studies by Mastitskaya et al. is that GLP-1R – mediated coronary vasodilatation involves fine myocardial capillaries covered by NG2+ pericytes (confocal microscopy studies on cardiac sections, ex vivo). These results are in line with recent studies in brain microcirculation (Chen Y, et al. Exendin-4 improves cerebral ischemia by relaxing microvessels, rapidly increasing cerebral blood flow after reperfusion. *Basic Res Cardiol.* 2025;120:423-441. doi: 10.1007/s00395-025-01096-y).

In the 2nd part of the manuscript Mastitskaya and colleagues performed pharmacological studies in isolated mouse right ventricular free wall preparations to dissect the intracellular mechanisms of the relaxing effects of GLP-1. Capillary diameters "at pericyte somata" were measured under ischemic conditions (mimicked by oxygen and glucose deprivation) or after infusion of endothelin-1, a potent microvascular constrictor (2-photon microscopy studies). The vasodilating effect of the GLP-1R agonist Exendin-4 was studied in the presence of specific activators/inhibitors of ATP-sensitive potassium channels (K_{ATP}) or AMPK inhibitors. The design of these experiments was based on previously published studies showing that GLP-

1 stimulates KATP-channels in vascular smooth muscle cells, as one mechanism mediating its vasodilating effect (Green BD, et al. GLP-1 and related peptides cause concentration-dependent relaxation of rat aorta through a pathway involving KATP and cAMP. Arch Biochem Biophys. 2008;478:136-42). In line with these previous observations, the present studies show that GLP-1R – mediated coronary vasodilatation involves the hyperpolarizing activation of KATP channels and the membrane insertion of these channels triggered by AMPK. Here, the involvement of pericytes is demonstrated by using right ventricles from mice with Kir6.1 deletion in pericytes (Kir6.1 is a subunit of the KATP channel).

Overall, this study is potentially interesting, although several main informations were already published before by other groups (the GLP-1/GLP-1R pathway participates in protective heart and brain ischemic preconditioning; in vascular cells (and also in cardiomyocytes) this pathway involves the activation of KATP channels).

For this reviewer, the most interesting part of the manuscript would be the clear demonstration of a brain-gut-cardiac pericytes axis (as stated in the manuscript title), but this axis is not clear. My major comments are:

- The manuscript (and previous studies) involve GLP-1 as endocrine circulating hormone. How are the plasma GLP-1 levels in rats/mice after ischemic preconditioning? Are they higher as for instance after a carbohydrate rich meal?!
- Are intestinal enteroendocrine K cells the only cells involved as source of GLP-1? (GLP-1 is also produced in other locations)
- To what extent is the myocardial “capillary dilatation at pericyte sites” dependent on the vasodilatation and enhanced flow through the feeding proximal arteries? (the manuscript repeatedly mentions “pericyte relaxation” but only shows the diameter of capillaries).
- A more direct and clear “dissection of the role of pericytes in cardiac “no-reflow” and ischemic preconditioning could be achieved by generating mice with pericyte-restricted GLP-1R deletion (floxed mice are available; see Kajitani Y, et al. High frequency of germline recombination in Nestin-Cre transgenic mice crossed with Glucagon-like peptide 1 receptor floxed mice. PLoS One. 2023;18:e0296006. doi: 10.1371/journal.pone.0296006). Intravenous, systemic delivery of Ex9 of course blocked GLP-R1 on other types of cells (in and outside the heart), which might be involved in the regulation of coronary perfusion.
- In figure 1C, the GLP-1R seems to be mostly in the cytosol of pericytes!
- Finally, it is not clear to this reviewer how/why the GLP-1/GLP-1R pathway activates KATP channels in VSMCs, pericytes (present studies) and cardiomyocytes, while blocking these channels in pancreatic beta cells (Leech CA, et al. Signal transduction of PACAP and GLP-1 in pancreatic beta cells. Ann N Y Acad Sci. 1996;805:81-92). The reference 19 cited in the manuscript in fact describes that the GLP-1/GLP-1R pathway normally blocks KATP channels in beta cells (as is physiologically meaningful), which was only reverted into an “activation” under very artificial conditions.

Reviewer #3

(Remarks to the Author)

The authors provide an interesting set of experiments utilising immunofluorescence labelling, antagonists and KO mice to explore the mainly sub cellular signalling pathways that are involved in pericyte constriction mediating no reflow in response to ischaemia (or OGD). They propose that the brain-gut-heart axis via vagally mediated GLP-1 release is important in signalling RIPc protective vasodilatory signals to the heart pericyte with sub cellular signalling that are dependent on KATP channel membrane expression / opening via a AMP kinase dependent signalling - that result in relaxation.

No reflow is an important clinical phenomenon and further understanding the pathophysiology is needed to prevent it occurring and to discover new treatments

General comments:

1. The experiments are elegantly demonstrated and illustrated with compelling results. However, the focus is mainly on sub cellular signalling which has been previously demonstrated - see work by Kloner RA and others. However, the data on pericytes are novel and thoroughly explored with a logical series of experiments.
2. The title of the paper and Figure 5a propose the brain-gut-heart signalling, although this was not specifically assessed in whole animal experiments and therefore possibly the title and conclusions drawn in the discussion over-reach, based on the actual data provided. Further whole animal experiments with possible nerve truncation or in vivo blood sampling for GLP-1 would strengthen the manuscript and support the brain-gut-heart signalling hypothesis.
3. GLP-1 is inherently unstable with a 2min half life due to degradation by DPP-4. Have the authors looked at potentiating the effect with DPP-4 inhibition, rather than only blocking it? How did they know that local GLP-1 released from the gut reaches the heart?
4. The Kir6.1 trafficking immunofluorescence intensity profiles measured using imageJ (Figure 4) needs further explanation/ verification with control studies. The images of pericytes/ selection of membrane were presumably not random and are possibly open to bias.
5. The methodology has been adequately described and the reported. However, the assessment of pericyte "opening" is only done visually (microscopically) (%open/ measured diameters), rather than employing other methods e.g. haemodynamic assessment of resistance that could add weight to the authors findings. Furthermore, visual assessment again may be open to reporting error/ selection bias.
6. The translation of these studies to Man has been challenging - with RIPc not showing improvement in microcirculatory function and mixed results in cardio protection although GLP is a vasodilator. This dichotomy between animal studies and studies in Man deserves further attention in the discussion.

Minor

1. Clinically important no-reflow is relatively rare (2% rather than 50% as stated) after coronary stenting - please revise introduction

2. There are typos in Figure 4: OGG rather than OGD

Version 2:

Reviewer comments:

Reviewer #1

(Remarks to the Author)

No further comments.

Reviewer #2

(Remarks to the Author)

The authors answered all my questions. This clarified which aspects of the manuscript are novel, and which parts mostly extend published data. Overall this is an interesting story.

As previously commented by reviewers 2 and 3, the title of the manuscript is not appropriate. The present studies do not further characterize an "endocrine axis", they provide novel mechanistic insights regarding the target cells and signalling pathways.

Reviewer #3

(Remarks to the Author)

The authors have addressed all my comments.

MS # NCOMMS-25-56046B

Responses to referees' comments

We are grateful for the constructive comments of the Reviewers on our original submission to the *Nature Communications* and have taken full account of the criticisms raised. We are delighted to have an opportunity to re-submit our work. We now provide a full response to the reviewers' comments as well as a thoroughly revised manuscript.

Below we state the criticisms ("critique") and then provide our responses.

REVIEWER COMMENTS

Reviewer #1

This an interesting and well-written manuscript providing evidence in support of the idea that remote preconditioning (RPC) protects the heart after ischemia via GLP-1 activation of KATP channels on pericytes, leading to capillary dilation and protection against the no-reflow phenomenon. The data appear to be of high quality and, at face value, provide compelling support. I have a few comments, some relatively minor.

Response: We would like to thank the reviewer for taking the time to review our manuscript and for their overall positive assessment of our work.

Critique 1:

Many channels are indirectly regulated by ATP. To be consistent with the literature, it would be better to use the generally accepted term: ATP-sensitive potassium channels, rather than ATP-gated.

Response: We thank the reviewer for pointing this out. We have now revised the text and use the term ATP-sensitive potassium channels throughout.

Critique 2:

Figs. 2b and d, do not look like blowups of a and c. Are the images inverted somehow?

Response: The areas outlined in red on panels a and c are just a schematic representation of the area used for high-power analysis of capillary blockages. The dimensions of the imaged area in 2b and 2d is only 309 μ m x 309 μ m while the area outlined in red on 2a and 2e is approximately 1.5mm x 1.5mm. We now revised the figure legend to describe this more precisely, as follows (page 2):

(a) Low-power view of heart cross-section after ischaemia/reperfusion (b) a zoomed-in image taken from part of the area outlined red in (a), with a line drawn across capillaries to calculate the ratio of blocked to total capillaries.

We also corrected the scale bar in figure 2c which should have been 50 μ m, not 20 μ m.

How was it decided where to draw the lines in Fig 2 b and d? Moving them up or down could give very different results in 2f.

Response: Three randomly selected zoomed-in images per heart were taken from the ischaemic area (part of which is outlined in red), with the capillaries running parallel to the surface of the tissue slice. A line perpendicular to the capillaries was drawn in the central area of each randomly placed image (± 50 μ m), the numbers of the blocked and perfused capillaries were calculated, and the percentage of the blocked capillaries was averaged per heart. The person performing the analysis was blinded to the condition.

This is now all stated on page 8. For the images mentioned by the referee, moving the line up or down by 50 μm would change the measured fraction of vessels blocked from 65% to 64% or 74% (for IR, Fig. 2b) or from 28% to 36-38% (for IR+RPC, Fig. 2e):

% of blocked capillaries on Fig.2b and 2e:

original values and if moving the line by $\sim 50\mu\text{m}$ from the center

	open (o)	blocked (x)	total	% blocked
Fig.2b (IR)	7	13	20	65
50 μm down	6	14	22	64
100 μm down	5	14	19	74
	open (o)	blocked (x)	total	% blocked
Fig.2e (IR+RPC)	13	5	18	28
50 μm up	9	5	14	36
50 μm down	13	8	21	38

Critique 3: Antibodies against KATP channel subunits have been notoriously unspecific. The authors could carry out control experiments on Kir6.1 knockout tissue to confirm the identified channels in Fig. 4.

Response: We performed all the necessary negative control staining and now provide the images of knockout tissue (Fig. 4d).

Critique 4: How strong is the evidence for the gut/CNS part of the proposed model in Fig. 5? GLP1 is synthesized in L-cells, but also in alpha cells. Could it be made elsewhere, and are the RPC effects lost with spinal lesion, for example?

Response: GLP-1 is indeed also produced in the alpha-cells, in type II taste receptor cells (the GLP-1R is also there - thus, GLP-1 is supposed to modulate taste), in the olfactory bulb and in the PPG neurons in the brainstem, but most likely none of these sources provide quantitatively enough GLP-1 to increase (peripheral) circulating levels to the point that GLP-1 can have systemic effects. The centrally produced GLP-1 has only been shown to have effects within the CNS, mostly associated with appetite control (particularly satiation after large meals) as well as fluid homeostasis. The evidence for the vagally mediated release of GLP-1 from the gut stems from our previous publication (Mastitskaya et al., *PlosONE* 2016, doi: 10.1371/journal.pone.0150108) where we performed a series of selective transections of different branches of the vagus, and demonstrated that the posterior gastric branch of the subdiaphragmatic vagus is responsible for the release of the molecular factor mediating RPC. This is all stated in the Introduction to the paper on page 1 (right column, lines 49-53) and Discussion (lines 242-244 and 269-272) of the manuscript.

Reviewer #2

In the present study Mastitskaya and colleagues studied whether coronary microvascular pericytes participate in the previously reported protective vasodilating effects of “endogenous” glucagon-like peptide (GLP-1) following remote ischemic preconditioning. In analogy to these previous studies from Basalay MV, et al. (“Glucagon-like peptide-1 (GLP-1) mediates cardioprotection by remote ischemic conditioning”. *Cardiovasc Res.* 2016;112:669-676. doi: 10.1093/cvr/cvw216) here the authors used a rat model of myocardial infarction (coronary artery occlusion followed by reperfusion). In such anesthetized rats, remote ischemic preconditioning (R_{Pc}) was induced by 15 min occlusion of femoral arteries applied 25 min prior to the myocardial ischemia. The present studies reproduce the effects observed by Basalay et al. showing that R_{Pc} attenuates the ischemia-induced reduction of coronary flow, and that this cardioprotective vasodilating effect is attenuated following blockade of GLP-1 receptors (GLP-1R) by Exendin(9-39) (Ex9, administered through systemic i.v. injection before LAD ligation). The novel aspect of the 1st part of the present studies by Mastitskaya et al. is that GLP-1R – mediated coronary vasodilatation involves fine myocardial capillaries covered by NG2⁺ pericytes (confocal microscopy studies on cardiac sections, ex vivo). These results are in line with recent studies in brain microcirculation (Chen Y, et al. Exendin-4 improves cerebral ischemia by relaxing microvessels, rapidly increasing cerebral blood flow after reperfusion. *Basic Res Cardiol.* 2025;120:423-441. doi: 10.1007/s00395-025-01096-y).

Response: In fact, the lead author of this paper was also second author on the Basalay et al. paper mentioned above, and this study is indeed a mechanistic analysis of how the effects of the Basalay et al. paper come about. We identify the cell type involved (coronary pericytes, as noted in the reviewer’s comments above), and also the ion channels involved as described below. Although the Chen et al. paper reports something similar for brain mural cells being able to restore blood flow after ischaemia, it does not specifically identify the underlying mechanism (attributing it to a guanylate cyclase independent NO effect and to PgE₂). We thus believe that our paper makes an important step forward, certainly for the heart, and in fact probably for pericytes in other vascular beds, thus opening up the possibility of developing a pharmacological approach to mimic the effects of remote ischaemic preconditioning.

In the 2nd part of the manuscript Mastitskaya and colleagues performed pharmacological studies in isolated mouse right ventricular free wall preparations to dissect the intracellular mechanisms of the relaxing effects of GLP-1. Capillary diameters “at pericyte somata” were measured under ischemic conditions (mimicked by oxygen and glucose deprivation) or after infusion of endothelin-1, a potent microvascular constrictor (2-photon microscopy studies). The vasodilating effect of the GLP-1R agonist Exendin-4 was studied in the presence of specific activators/inhibitors of ATP-sensitive potassium channels (K_{ATP}) or AMPK inhibitors. The design of these experiments was based on previously published studies showing that GLP-1 stimulates K_{ATP}-channels in vascular smooth muscle cells, as one mechanism mediating its vasodilating effect (Green BD, et al. GLP-1 and related peptides cause concentration-dependent relaxation of rat aorta through a pathway involving K_{ATP} and cAMP. *Arch Biochem Biophys.* 2008;478:136-42). In line with these previous observations, the present studies show that GLP-1R – mediated coronary vasodilatation involves the hyperpolarizing activation of K_{ATP} channels and the membrane insertion of these channels triggered by AMPK. Here, the involvement of pericytes is demonstrated by using right ventricles from mice with Kir6.1 deletion in pericytes (Kir6.1 is a subunit of the K_{ATP} channel).

Response: As the referee notes, we show that restoration of coronary blood flow by remote ischaemic preconditioning is mediated by GLP-1 activating ATP-sensitive K

channels, and we show how this depends crucially on signalling pathways that insert these channels into the cell membrane. However, this goes far beyond the Green et al. paper mentioned by the referee (which we do cite in the paper), through establishing:

- (i) the role of GLP-1 in mediating the preconditioning evoked coronary blood flow increase;
- (ii) the role of GLP-1 in coronary capillaries (as opposed to arteries or arterioles) which are the part of the coronary circulation that cause no-reflow after cardiac ischaemia, thus opening up the possibility for developing a pharmacological approach to mimic the effects of remote ischaemic preconditioning;
- (iii) that the GLP1R antagonist Exendin(9-39) has an effect opposite to the agonist Exendin-4 (unlike in the Green et al. paper on the aorta);
- (iv) the modulatory role of other signalling pathways that regulate K_{ATP} channel trafficking.

Thus, the impression given by the referee's comments, that our work is not novel, is highly misleading.

Overall, this study is potentially interesting, although several main informations were already published before by other groups (the GLP-1/GLP-1R pathway participates in protective heart and brain ischemic preconditioning; in vascular cells (and also in cardiomyocytes) this pathway involves the activation of K_{ATP} channels).

Response: We would like to thank the reviewer for their overall positive assessment of our work. However, we believe that the major advances described in this manuscript, viz. discovering the mechanism of remote ischaemic preconditioning, identifying the receptors and ion channels involved, and defining how this mechanism is regulated by other messenger systems, make it far more novel and interesting than the referee is stating.

For this reviewer, the most interesting part of the manuscript would be the clear demonstration of a brain-gut-cardiac pericytes axis (as stated in the manuscript title), but this axis is not clear.

Response: As noted above for Referee 1, the evidence for the vagally mediated release of GLP-1 from the gut is provided in our previous publication (Mastitskaya et al, *PlosONE* 2016, doi: 10.1371/journal.pone.0150108), where we performed a series of selective transections of different branches of the vagus, and demonstrated that the posterior gastric branch of the subdiaphragmatic vagus is responsible for the release of the molecular factor mediating RPC.

My major comments are:

Critique 1: The manuscript (and previous studies) involve GLP-1 as endocrine circulating hormone. How are the plasma GLP-1 levels in rats/mice after ischemic preconditioning? Are they higher as for instance after a carbohydrate rich meal?!

Response: We did not measure plasma GLP-1 levels in this study, however, we published data on [GLP-1] previously in Basalay et al. (2016, doi: 10.1093/cvr/cvw216). There we observed a moderate GLP-1 level increase by 1.7 pg/ml (or 0.52 pM) in arterial blood 30 min after the RPC stimulus. For comparison, the absolute increase in total GLP-1 in plasma from the retrobulbar plexus (containing both arterial and venous blood) was only ~0.5 pM in mice 5 min after oral glucose – a value similar to our findings on RPC effects on plasma levels - and declined below the baseline by 20 min (see Fig 2A-B of Smits et al, 2024, reproduced below).

GLP-1 measurements are notoriously difficult and usually have a large variability – a recent study even indicates that accurate measurements of active GLP-1 in the blood are only possible with in vivo blockade of the degrading enzyme DPPIV (Smits et al, 2024 doi: 10.2337/db23-0848). Thus, adding DPPIV inhibitors in vivo prior to giving oral glucose resulted in glucose evoking a temporary 6pM GLP-1 increase (see Fig 4E of Smits et al, 2024 doi: 10.2337/db23-0848), reproduced below: intact [unmetabolised] GLP-1 levels measured after giving glucose at t=0 are shown in control conditions without peptidase inhibition (black dashed line, and in the presence of the DPPIV inhibitor valine pyrrolidide or of a combination of valine pyrrolidide with the neprilysin inhibitor sacubitril (given orally 30 min prior to the glucose) which are shown as the purple dashed line and the solid purple line respectively.

Fig 4E of Smits et al, 2024 (relabelled for easier understanding)

Critique 2: Are intestinal enteroendocrine K cells the only cells involved as source of GLP-1? (GLP-1 is also produced in other locations)

Response: We believe there is a typo here, and the Reviewer means L-cells. As per our response to the similar comment from Reviewer #1, GLP-1 is also produced in the alpha-cells, in type II taste receptor cells, in the olfactory bulb and in the PPG neurons in the brainstem, but most likely none of these sources provide quantitatively enough GLP-1 to increase (peripheral) circulating levels to the point that GLP-1 can have systemic effects.

The centrally produced GLP-1 has only been shown to have effects within the CNS, mostly associated with appetite control and particularly satiation after large meals as well as fluid homeostasis. The evidence for the vagally mediated release of GLP-1 from the gut being the major contributor to cardioprotective effects of RPc is given in our previous publication (Mastitskaya et al, PlosONE 2016, doi: 10.1371/journal.pone.0150108) where we performed a series of selective transections of different branches of the vagus, and demonstrated that the posterior gastric branch of the subdiaphragmatic vagus is responsible for the release of the molecular factor mediating RPc.

Critique 3: To what extent is the myocardial “capillary dilatation at pericyte sites” dependent on the vasodilatation and enhanced flow through the feeding proximal arteries? (the manuscript repeatedly mentions “pericyte relaxation” but only shows the diameter of capillaries).

Response: Pericytes are the only contractile cells on capillaries (Peppiatt et al., 2006, *Nature* doi.org/10.1038/nature05193), and if the capillary diameter decreases or increases “actively” it is due to pericyte contraction or relaxation. We have previously demonstrated for the brain that pericyte relaxation in response to neural activity occurs prior to changes of the diameter of feeding arteriole (Hall et al., 2014, *Nature*, doi.org/10.1038/nature13165), and that the resulting dilation occurs mostly near the cell soma where most of the pericyte’s circumferential processes are located (Nortley et al., 2019, *Science* doi.org/10.1126/science.aav9518). For the brain we estimated (based on detailed measurements of the topology of the vascular network by David Kleinfeld’s group, doi.org/10.1038/nn.3426) that capillary pericytes are responsible for ~80% of the increase in blood flow that occurs in response to neuronal activity (Hall et al., 2014, *Nature*, doi.org/10.1038/nature13165). A similar topological and quantitative analysis has yet to be performed for the coronary vasculature. The point, however, is that pericytes can actively relax to increase blood flow, and capillaries are not just being pushed open as a result of upstream arteriole dilation (Hall et al., 2014, *Nature*, doi.org/10.1038/nature13165, see Figure 3).

To explain why we measure capillary diameters at the somata of pericytes, we have now added to the manuscript the figure below (as Fig. 3b) plotting the capillary diameter changes in the cardiac microvasculature as a function of distance from pericyte somata, in response to the vasoconstrictor endothelin-1. As in the brain, the diameter change is focused near the pericyte somata, due to most circumferential processes (which have the optimal orientation to change the diameter) being located there (see Supplementary Figs 2 and 3 of Nortley et al., 2019, *Science* doi.org/10.1126/science.aav9518).

Critique 4: A more direct and clear “dissection of the role of pericytes in cardiac “no-reflow” and ischemic preconditioning could be achieved by generating mice with pericyte-restricted GLP-1R deletion (floxed mice are available; see Kajitani Y, et al. High frequency of germline recombination in Nestin-Cre transgenic mice crossed with Glucagon-like peptide 1 receptor floxed mice. PLoS One. 2023;18:e0296006. doi: 10.1371/journal.pone.0296006). Intravenous, systemic delivery of Ex9 of course blocked GLP-R1 on other types of cells (in and outside the heart), which might be involved in the regulation of coronary perfusion.

Response: Thank you, this is an excellent idea, and we are considering doing it in the future. However, to do the experiment we would need both the floxed mouse mentioned, and a truly pericyte-specific Cre driver line. The problems of obtaining a truly pericyte-specific Cre driver have been described by Guo et al., (2024, <https://pmc.ncbi.nlm.nih.gov/articles/PMC11502228/>). The commonly used NG2-Cre line can delete floxed proteins from pericytes but also does so in some smooth muscle cells and in oligodendrocyte precursor cells. Although there is a mouse driver line that is truly specific for pericytes in the brain (Guo et al., cited above) no such line exists for coronary capillary pericytes. Furthermore, even if we used the NG2-Cre mouse, because we currently do not have the floxed GLP-1R mouse, the 6 months needed for mouse import, quarantine and breeding with the Cre mouse make it unfeasible to complete these experiments for this paper in a reasonable timeframe.

We would, however, like to point out that we can be absolutely sure that the Exendin-4 is affecting the contractile properties of coronary pericytes, because we are directly imaging the changes of capillary diameter produced by the pericytes. Furthermore, we have now added to the manuscript a series of additional live tissue imaging experiments, using a similar approach to that in Figure 3c, employing conditions of oxygen-glucose deprivation to constrict capillaries, and application of Ex4 to activate capillary GLP-1Rs; but instead of blocking KATP channels with glibenclamide, we now blocked GLP-1Rs with Exendin(9-39). The results are now included as Figure 3d, and show that the dilating effect of the GLP-1R mimetic Exendin-4 is blocked (as expected) by the GLP-1R antagonist Exendin(9-39):

Critique 5: In figure 1C, the GLP-1R seems to be mostly in the cytosol of pericytes!

Response: For Figure 1C, we used Fluorescein-Trp25-Exendin-4 (FLEX) to label the GLP-1 receptors. This is a fluorescently tagged Exendin-4, i.e. a GLP-1R agonist, and so internalisation of the FLEX-GLP-1R complex (and hence a cytosolic appearance of the labelling) is not surprising (and has been reported previously by Roed SN et al, 2014 doi: 10.1016/j.mce.2013.11.010).

Critique 6: Finally, it is not clear to this reviewer how/why the GLP-1/GLP-1R pathway activates K_{ATP} channels in VSMCs, pericytes (present studies) and cardiomyocytes, while blocking these channels in pancreatic beta cells (Leech CA, et al. Signal transduction of PACAP and GLP-1 in pancreatic beta cells. *Ann N Y Acad Sci.* 1996;805:81-92). The reference 19 cited in the manuscript in fact describes that the GLP-1/GLP-1R pathway normally blocks K_{ATP} channels in beta cells (as is physiologically meaningful), which was only reverted into an "activation" under very artificial conditions.

Response: We are aware of this apparent discrepancy, but it can easily be explained by the different mechanisms involved. Although this is not the focus of our paper, which is on how remote ischaemic preconditioning is mediated by GLP-1 activating K_{ATP} channels in pericytes, GLP-1 potentiates the closing of K_{ATP} channels in beta-cells for the following reasons. Canonically, a rise of glucose concentration tends to close K_{ATP} channels because it raises intracellular [ATP] (McTaggart et al, *J Physiology* 2010. doi: 10.1113/jphysiol.2010.191767). This effect is potentiated by GLP-1R activation as a result of downstream activation of PKA and hence EPAC (Exchange Protein directly Activated by Cyclic AMP) (Marzook et al, *Front Endocrinol* 2021 doi: 10.3389/fendo.2021.678055). EPAC increases the sensitivity of K(ATP) to ATP and hydrolyses PIP2 which tends to keep K_{ATP} channels open (Kang et al, *J Physiol.* 2006 doi: 10.1113/jphysiol.2006.107391; Baukowitz et al. *Science* 1998 doi: 10.1126/science.282.5391.1141). Without the initial glucose-dependent step, GLP-1 cannot exert its potentiating effect on K_{ATP} channel closure in beta-cells. Therefore, without the presence of glucose, GLP-1 has only negligible effects on insulin release, which is why GLP-1-based therapies are so useful for the treatment of Type 2 diabetes.

The K_{ATP} channel in beta-cells is made of Sur1 and Kir6.2 subunits. PKA phosphorylates the SUR1 subunit which leads to a decrease in the channel's opening probability, enhances sensitivity to ATP and inhibits the counteracting (opening) effect of ADP.

In contrast, in pericytes, K_{ATP} channels are made of Sur2B and Kir6.1 subunits. The downstream effect of PKA activation on K_{ATP} channels for these cells differs from in beta cells. PKA phosphorylation of Sur2B activates (opens) K_{ATP} channels and is, for example, responsible for beta2-adrenergic effects on vascular relaxation (Shi Y et al, 2007 doi: 10.1152/ajpregu.00337.2007). PKA can also phosphorylate multiple sites of Sur2B leading to vasorelaxation (Quinn KV et al, 2004, doi: 10.1161/01.RES.0000128513.34817.c4).

Reviewer #3 (Remarks to the Author):

The authors provide an interesting set of experiments utilising immunofluorescence labelling, antagonists and KO mice to explore the mainly sub cellular signalling pathways that are involved in pericyte constriction mediating no reflow in response to ischaemia (or OGD). They propose that the brain-gut-heart axis via vagally mediated GLP-1 release is important in signalling RPc protective vasodilatory signals to the heart pericyte with sub cellular signalling that are dependent on KATP channel membrane expression / opening via a AMP kinase dependent signalling - that result in relaxation.

No reflow is an important clinical phenomenon and further understanding the pathophysiology is needed to prevent it occurring and to discover new treatments.

We thank the referee for these positive comments.

General comments:

Critique 1: The experiments are elegantly demonstrated and illustrated with compelling results. However, the focus is mainly on sub cellular signalling which has been previously demonstrated - see work by Kloner RA and others. However, the data on pericytes are novel and thoroughly explored with a logical series of experiments.

Response: We thank the referee for their positive comment on the pericyte aspect of our work. As far as the supposedly previous demonstration of the subcellular signalling is concerned, unfortunately, the referee cites no papers, and in PubMed we could find only one editorial (and no original papers) on GLP-1 by Kloner R.A., and a single paper discounting *mitochondrial* K_{ATP} channels as modulating myocardial infarct size. As far as we are aware, our discoveries on the mechanism of how RPc affects the coronary microvasculature after ischaemia have not been reported previously.

Critique 2: The title of the paper and Figure 5a propose the brain-gut-heart signalling, although this was not specifically assessed in whole animal experiments and therefore possibly the title and conclusions drawn in the discussion over-reach, based on the actual data provided. Further whole animal experiments with possible nerve truncation or in vivo blood sampling for GLP-1 would strengthen the manuscript and support the brain-gut-heart signalling hypothesis.

Response: Thank you for this comment and the suggestion of the experiments for demonstration of the brain-gut-heart signaling hypothesis. These experiments were performed in our previous studies on cardioprotective effects of GLP-1. Firstly, in a study on selectively transecting different branches of the vagus (Mastitskaya et al, *PlosONE* 2016, <https://doi.org/10.1371/journal.pone.0150108>), we demonstrated that the subdiaphragmatic posterior branch of the vagus nerve is responsible for RPc-induced cardioprotection (the sectioning of this branch prevented cardioprotection, while selective electrical stimulation of it resulted in reduced infarct size; with all appropriate controls in place). Secondly, we sampled the blood *in vivo* in the rats undergoing temporary limb ischaemia (the equivalent of RPc) and showed that after 30 min of ischaemia onset, the plasma levels of GLP-1 were elevated (Basalay et al, 2016 *Cardiovascular Res* doi: 10.1093/cvr/cvw216). In the current paper, we demonstrate in experiments *in vivo* that GLP-1R antagonists prevent the protective effects of RPc on no-reflow in rats (which supports the findings in the Basalay et al, 2016 study), and that protective effects of RPc are absent in mice with selective Kir6.1 KO in pericytes.

Our earlier experiments establishing a role for the brain-gut-heart aspect of RPC were described in the Introduction to the paper on page 1 (right column, lines 49-53) of the manuscript.

Critique 3: GLP-1 is inherently unstable with a 2min half life due to degradation by DPP-4. Have the authors looked at potentiating the effect with DPP-4 inhibition, rather than only blocking it? How did they know that local GLP-1 released from the gut reaches the heart?

Response: We deduce that the cardioprotection conferred by RPC is mediated by GLP-1 reaching the heart because it is prevented by the GLP-1R antagonist Exendin(9-39) (Fig. 2c, f of the manuscript).

We have not tested the effect of DPP4 inhibition because it has some effects that are not mediated by the GLP-1 system (including via GIP, neuropeptide Y and SDF-1 α). Indeed, it is difficult to distinguish the cardioprotective effects of DPP4 inhibitors from their effects mediated by increased levels of GLP-1. It is well established that DPP4 inhibitors have prosurvival effects on myocardium via activation of a SDF-1 α /CXCR4-mediated STAT3 prosurvival pathway, dramatically reducing the infarct size (Kubota et al, 2016 doi: 10.1016/j.yjmcc.2015.12.026), while GLP-1R signaling activates a PI3K/Akt prosurvival pathway in myocardium.

Critique 4: The Kir6.1 trafficking immunofluorescence intensity profiles measured using imageJ (Figure 4) needs further explanation/ verification with control studies. The images of pericytes/ selection of membrane were presumably not random and are possibly open to bias.

Response: As stated in the Methods, the person analysing the images was blinded to the experimental condition, with the aim of removing possible operator bias. Between 3-5 zoomed-in images per heart were taken to achieve a total number of 5-10 pericytes per heart, and all imaged pericytes were analysed. The selection of the membrane area within each cell was based on having a clearly visible PDGFR β signal, on the outer side of the capillary, as has now been described in the Methods (lines 647-651).

Critique 5: The methodology has been adequately described and the reported. However, the assessment of pericyte "opening" is only done visually (microscopically) (%open/ measured diameters), rather than employing other methods e.g. haemodynamic assessment of resistance that could add weight to the authors findings. Furthermore, visual assessment again may be open to reporting error/ selection bias.

Response: There is a strong correlation between the diameter measured at the pericytes in Figures 2, 3 and 4, and the amount of tissue perfusion in Figure 2, strongly suggesting that it is the pericyte-mediated capillary constriction (or its absence) that restricts (or allows) coronary blood flow. It would be ideal if a computer model of the coronary vasculature could be used to convert our measurements of diameter changes at pericytes into a change of coronary blood flow, but such a model does not yet exist (unlike for the brain where capillaries are reported to confer 4 times the resistance conferred by arterioles to the flow of blood: Gould et al., 2017, JCBFM, <https://doi.org/10.1177/0271678x16671146>). Measurement of capillary diameters was performed by an observer blinded to the condition being applied and diameters at all pericyte locations in the images were measured.

Critique 6: The translation of these studies to Man has been challenging - with RIPc not showing improvement in microcirculatory function and mixed results in cardio protection although GLP is a vasodilator. This dichotomy between animal studies and studies in Man deserves further attention in the discussion.

Response: Thank you for this comment. We now added a commentary on this issue to the Discussion (lines 352-379). Indeed, the translation of RPc cardioprotective studies has been challenging. Partly, the enthusiasm in the field was significantly compromised by the poor design of some major clinical trials on RPc, e.g. where RPc was performed in patients undergoing PCI procedures and the anaesthesia included propofol which is known to suppress vagal tone. Given that the vagal pathway is crucial in mediating cardioprotection, it is not surprising that the studies performed under propofol anaesthesia resulted in neutral outcomes.

More importantly, our aim, in defining the mechanism by which RPc is cardioprotective, is not to advocate RPc for clinical use, but to identify a signalling mechanism that can be targeted to confer cardioprotection without the need for RPc.

Minor:

1. Clinically important no-reflow is relatively rare (2% rather than 50% as stated) after coronary stenting - please revise introduction.

Response: The Reviewer cites no reference but is apparently referring to the statistics on no-reflow visible on angiography. Indeed, the no-reflow of major coronary blood vessels is relatively rare. However, we are providing the statistics for microvascular no-reflow, as discussed in numerous reviews (doi: 10.1055/s-0041-1725979; doi: 10.1016/j.yjmcc.2011.06.009).

2. There are typos in Figure 4: OGG rather than OGD.

Response: Thank you. We have corrected this.